# Energy-Efficient BWP Configuration for Multi-Slice Users

**DOI:** 10.3390/s24041281

**Published:** 2024-02-17

**Authors:** Joe Saad, Kinda Khawam, Mohamad Yassin, Salvatore Costanzo

**Affiliations:** 1Orange Innovation, 44 Avenue de la République, 92320 Châtillon, France; joe.saad@orange.com (J.S.); mohamad.yassin@orange.com (M.Y.); salvatore.costanzo@orange.com (S.C.); 2Laboratoire DAVID, UFR des Sciences, University of Versailles, 45 Avenue des Etats-Unis, 78000 Versailles, France

**Keywords:** 5G, energy efficiency, QoS, multi-slice connectivity, BWP

## Abstract

Fifth Generation (5G) mobile networks introduce the concept of slicing to ensure isolation among the various supported heterogeneous services. The User Equipment (UE) can be connected to multiple slices simultaneously. Additionally, the notion of a Bandwidth Part (BWP) was also instigated to reduce power consumption. A BWP is a small chunk of the bandwidth scanned by the UE to retrieve its service data. Therefore, a UE connected to multiple services can be configured with multiple BWPs each associated with a given service. Such UEs find themselves scanning multiple BWPs, which can be time consuming and highly energy intensive. Hence, it is paramount to study the appropriate choice of the BWP configuration from an energy-efficiency perspective for multi-slice users depending on their battery level. In this paper, two energy-efficient BWP selection solutions are proposed for users connected to multiple slices. The first solution is based on a centralized approach where UEs are stirred optimally to the best BWP configuration, while the second solution relies on a user-centric distributed approach using non-cooperative game theory. The proposed schemes take into account the users’ battery level and their sojourn time in the network as well as the scanned BWP size. Both solutions are compared with one another and against the legacy solution. Intensive simulation results demonstrate the efficiency of our proposition in terms of users’ energy efficiency and quality of service.

## 1. Introduction

Fifth Generation (5G) mobile networks support multiple heterogeneous services including the enhanced Mobile Broadband (eMBB) service requiring high throughput demand and the Ultra-Reliable Low-Latency Communications (URLLC) service requiring low latency and high reliability [1]. Owing to the concept of slicing, the physical network is partitioned into multiple logical networks (coined slices) where each slice is dedicated to a service and has a dedicated number of radio resources. User Equipment (UE) can be connected with up to 8 slices simultaneously [2].

Moreover, the concept of a Bandwidth Part (BWP) was introduced in 5G to reduce UE power consumption where the UE can scan a limited part of the carrier band, termed BWP, instead of scanning the whole band. Additionally, this concept also supports flexible SubCarrier Spacing (SCS) designated by numerologies, where higher numerologies ensure lower latency and can be dedicated to more delay stringent services [3]. As such, a BWP is a set of contiguous Physical Resource Blocks (PRBs) linked to a specific numerology.

In this work, UEs connected to two slices (eMBB and URLLC) are considered where each service requires a different numerology. Hence, for these users, two options are envisaged. The first consists of attributing different BWPs with different numerologies for each service (or slice) adequately. In such a case, BWP switching is required to retrieve the service-related data of each BWP. Hence, although this option endows each service with the most appropriate Quality of Service (QoS), it increases the UE power consumption due to scanning a wider bandwidth, and it inflicts an additional latency due to the BWP switching delay and signaling messages exchanged to perform BWP switching [4,5]. The second option is to use a single BWP for all slices. Such an option corresponds to the legacy scheme and is applied currently in 5G networks. It has the merit to reduce complex signaling messages, costly BWP scanning and UE energy consumption but at the cost of reduced QoS. In fact, resorting to a single numerology (single BWP) fails to ensure satisfactory QoS for demanding services. Furthermore, using a single BWP with a single numerology may increase the UE’s sojourn time in the network, which in turn increases its energy consumption. Hence, we devise a sagacious and flexible scheme that selects the most appropriate BWP configuration for each UE among the two previously mentioned solutions: either the multi-numerology BWPs per UE slice or a single-numerology BWP for all the UE slices, depending on the UE battery level and QoS strictness. To reach that aim, two approaches are adopted: a distributed approach using non-cooperative game theory where every UE autonomously selects the BWP configuration (multi-numerology or single numerology) that strikes a good balance between improving its QoS and reducing its battery consumption and a centralized approach where UEs are assigned optimally to the most adequate BWP configuration. In both approaches, a cost function is carefully defined and takes into account the BWP size, the BWP level of congestion, the UE sojourn time in the network and the UE battery level. The performances of both approaches are assessed through extensive simulations where they are compared against each other and against the legacy scheme which they largely surpassed. The rest of the paper is organized as follows. Section 2 gives the state of the art. Section 3 discusses our system model. Section 4 and Section 5 tackle the proposed distributed and centralized approaches. Section 6 provides simulation results of the proposed solutions. Finally, Section 7 concludes the paper.

## 2. Related Works

In the state of the art (SOTA), most works focus on the dynamic allocation of the slice radio resources for users connected to a single slice such as our previous work [6] and the work in [7]. However, these works do not consider the full energy efficiency aspect nor the users’ multi-user-slice connectivity, which is the case of our work. In fact, few works such as the work in [8] consider users connected to multiple slices. In the latter, the authors used Deep Reinforcement Learning (DRL) to attribute the radio resources for each service to these users connected to multiple slices. Nevertheless, the radio resource allocation problem is addressed from a PRB allocation perspective, and the energy efficiency aspect is not taken into account. This is different from our work where the radio resources are allocated based on an energy-efficient BWP configuration selection. Other works from SOTA tackle the BWP subject by studying the BWP switching process such as the work in [9], where the impact of BWP switching on the network’s performance is assessed. Moreover, BWP switching related to the BWP inactivity timer is addressed in the work in [10], where a new method is proposed to manage the BWP Inactivity Timer by reducing latency and increasing throughput. Also, the authors in [11] propose an energy-efficient joint mechanism that combines BWP and Discontinous Reception (DRX). However, our work relies on a BWP configuration selection to reduce energy consumption instead of using existing methods such as DRX and the BWP Inactivity Timer. In addition, the mentioned works do not consider users connected to multiple slices.

For the energy-efficiency aspect, the work in [12] provides an overview of new features in 5G which help reduce power consumption and increase energy efficiency. In addition, some works propose energy-efficient virtual resource allocation methods using Deep Reinforcement Learning such as the work in [13]. Other works propose an energy-efficient routing protocol in 5G such as the work in [14], whereas the authors in [15] introduce an energy-efficient scheme by solving an optimization problem that aims to reduce power consumption. Nevertheless, these works do not tackle the energy-efficient aspect from a BWP radio resource allocation perspective and do not consider the multi-slice users.

In fact, the main focus of our work is to optimize the energy efficiency for users connected to multiple slices in order to ensure a balance between QoS satisfaction and energy efficiency, which is rarely considered in the SOTA. In fact, for these users, two options are available. The first is using a single numerology and a single BWP for all services and slices. With this option, energy consumption is reduced since BWP switching is avoided as well as scanning a larger bandwidth. However, it is not optimal to use a single BWP for all services, as each service may require the use of different numerologies for QoS performance satisfaction. For example, the URLLC service typically requires a higher numerology than the eMBB service [16]. The second option would be to use different BWPs and numerologies for each service, which helps to optimize the services’ QoS but at the cost of increasing energy consumption and inflicting additional delays due to BWP switching. Hence, a satisfactory compromise is needed between QoS satisfaction and energy efficiency. For this reason, we consider a scheme that flexibly selects one of these two options for each UE depending on the UE characteristics, since resorting to a single choice is detrimental to UE performance.

In fact, opting for a single numerology will reduce the energy consumption of the UE relative to BWP scanning and BWP switching. Nonetheless, a single numerology bestows lower data rates on the UE, which increases its sojourn time, leading to higher energy consumption. Conversely, a multi-numerology reduces the UE sojourn time owing to higher QoS satisfaction and data rates but in turn increases energy consumption because of costly BWP scanning and switching. Hence, this work tackles the challenging problem of energy-efficient BWP configuration selection for users connected to multiple slices.

The BWP configuration selection between a multi-numerology and single numerology is assessed for each user depending on multiple factors.

The devised BWP configuration selection aims to strike a good balance between the size of the scanned BWP and the UE sojourn time in the network depending on the UE QoS and UE battery level. Two approaches are adopted:A distributed approach based on a non-cooperative congestion game.A centralized approach based on a global optimization problem.

These approaches are compared against one another and against the legacy scheme where a single BWP for all services is selected for all users.

Thus, the contributions of our work are summarized as follows:Users connected to multiple slices are considered, which are rarely tackled in the SOTA.The selection of a BWP configuration for multi-slice users is used to address the radio resource allocation problem unlike other works from the literature focusing on the PRB allocation.This BWP configuration scheme is based on a novel concept where either the multi-numerology or single-numerology BWP configuration is selected for each multi-slice user while aiming to optimize the users’ energy consumption and QoS.The energy-efficiency aspect is considered in the BWP configuration selection process instead of using existing mechanisms from the standards such as DRX and the BWP Inactivity Timer to reduce users’ energy consumption.A centralized and a distributed approach are proposed and are compared against each other and against the legacy scheme.

The proposed schemes are detailed in the next sections.

## 3. System Model

The system model is represented in Figure 1.

As can be seen, we consider a fixed random number of users $N_{users}$ and a single next generation NodeB (gNB) with a total band $B_{total}$ which corresponds to the amount of bandwidth owned by a 5G operator. The gNB covers an area with radius *R*. Additionally, it operates on the $f_{operator}=\;3.5$ GHz frequency (Frequency Range 1) in Time-Division Duplex (TDD) mode. As for the users, they are randomly distributed in the gNB coverage area within the radius *R* where each user is connected to two slices: eMBB and URLLC. Each slice will have a dedicated number of radio resources. Additionally, three bandwidth parts are considered for each slice: the first BWP denoted by BWP 1 consists of the band attributed to the eMBB slice and uses numerology 1 (the lower numerology) with a band $B_{eMBB}$ MHz, the second BWP denoted by BWP 2 is the band attributed to the URLLC slice using numerology 2 (a higher numerology) with a band $B_{URLLC}$ MHz. Note that users selecting the multi-numerology BWP configuration will scan both these BWPs (BWP 1 and BWP 2) consecutively to retrieve the data for each slice. The third BWP denoted by BWP 3 is the one shared between both slices using numerology 1 with a band $B_{mixed}$ with $B_{eMBB}+B_{URLLC}+B_{mixed}\leq B_{total}$. This particular BWP (BWP 3) is scanned by users affected with a single numerology for both services. Each UE *u* has a given volume of data $V_{u,s}$ to retrieve for each service *s* and remain in the network until the total data volume for both services is consumed. A fair queuing scheduling is applied at the level of each BWP where the BWP radio resources are attributed equally among users. For each user, we calculate the throughput to determine at a later stage the user’s sojourn time in the network. The throughput of UE *u* for service *s* is computed as follows:(1)$$Thr_{u,s}=\frac{{AllocPRB}_{u,s}\times Bits\_per\_PRB_{u}}{D}$$
where ${AllocPRB}_{u,s}$ is the number of allocated PRBs to UE *u* for service *s* from its attached BWP during a TTI with duration *D*. Furthermore, $Bits\_per\_PRB_{u}$ is the number of bits per PRB that depends on the modulation and coding rate of the UE, which in turn are based on its Signal-to-Interference-plus-Noise Ratio (SINR). It is calculated as follows:(2)$$\begin{array}{c} Bits\_per\_PRB_{u}=12\times 14\times \log_{2}\left(MO_{u}\right)\times CR_{u} \end{array}$$

In (Equation 2), the PRB is considered to be composed of 12 subcarriers in the frequency domain and 14 OFDM symbols in the time domain, which explains the displayed values. In addition, $MO_{u}$ is the modulation order and $CR_{u}$ is the coding rate of the considered UE. As for the number of allocated PRBs to the UE for service *s*, it depends on the total number of PRBs ${TotPRB}_{bwp}$ allotted to its attached BWP bwp, which is linked to the service *s* and the total number of connected users to this particular BWP $N_{users,bwp}$, since a fair resource scheduling is applied:(3)$${AllocPRB}_{u,s}=\frac{{TotPRB}_{bwp}}{N_{users,bwp}}$$

As for the total number of PRBs available at each BWP, it is determined by Table 5.3.2-1 from [17] depending on the attributed BWP band and numerology.

Such performance indicators will be used to devise the cost function presented hereafter and used to stir adequately the BWP selection configuration in centralized and distributed approaches.

### The Cost Function

The cost function for UE *u* selecting BWP configuration strategy *c* is given by:(4)$$C_{u,c}=\sum_{s\in \{eb,uc\}}\alpha_{u,s}\cdot {DST}_{u,s,c}+\theta_{c}\cdot SwDl_{u}+\beta_{u}\cdot B_{c}$$
where

$DST_{u,s,c}$ is the sojourn time of UE *u* for retrieving the data of a particular service *s* when selecting strategy *c*. The service *s* can be either eMBB denoted by eb or URLLC denoted by uc.$SwDl_{u}$ is the BWP switching delay of UE *u* (also a part of the user’s overall sojourn time).$B_{c}$ is the total band scanned by the user depending on its devised strategy *c*.$\alpha_{u,s}$ and $\beta_{u}$ are normalizing factors. The former reflects the class of service *s* for UE *u* and the latter represents the battery level of UE *u*.$\theta_{c}$ is an indicator variable that equates to one in the presence of BWP switching delay and to zero otherwise, and hence it depends on the selected strategy.

In the cost function (Equation 4), the first term represents the UE’s sojourn time in the network, which is necessary for retrieving the data of all its services. That first term is used as a QoS indicator as the higher the user sojourn time, the lower its QoS, since the user will endure a higher delay and will stay active in the network longer, which may increase its energy consumption. Hence, a higher sojourn time increases the cost function. The second term is the BWP switching delay (when applicable), which is also a part of the user’s sojourn time in the network. The third term of the cost function represents the scanned BWPs by the UE. The higher this band, the higher the energy consumed by the user for scanning a larger band. In fact, if UE *u* chooses the multi-numerology strategy MN, it will scan consecutively BWP 1 and BWP 2 to retrieve the data of each service *s*, and the corresponding sojourn time for service *s* is determined by the following equation:(5)$$DST_{u,s,MN}=\frac{V_{u,s}}{Thr_{u,s}}$$
where

$V_{u,s}$ is the volume of data to be retrieved by UE *u* for service *s*.$Thr_{u,s}$ is the throughput of UE *u* for service *s*.

Additionally, with the multi-numerology scheme, the BWP switching delay is taken into account ($\theta_{MN}=1$) and the total scanned band by the user is the sum of BWP 1 and BWP 2 ($B_{MN}=B_{BWP1}+B_{BWP2}$). When UE *u* selects the single-numerology configuration SN, it will scan BWP 3 solely to retrieve the data of both services. Therefore, it will have the same achieved throughput for both services, since the same number of PRBs is allocated to the user for each service *s*. Therefore, $Thr_{u}$ is independent of the type of service in that case and $DST_{u,s,SN}=\frac{V_{u,s}}{Thr_{u}}$. Also, $\theta_{SN}=0$ and $B_{SN}=B_{BWP3}$ as only one BWP is scanned, which prevents BWP switching. Choosing the multi-numerology configuration will improve user’s KPIs and QoS and will increase user’s throughput, which in turn will reduce the user’s sojourn time (first term) but at the cost of adding a BWP switching delay (second term) and increasing the BWP scanned by the user (third term) and hence the ensuing energy consumption. In contrast, a single numerology avoids BWP switching (second term) and reduces the energy consumption, as the BWP scanned by the user (third term) is narrower but increases the user’s sojourn time (first term). Additionally, the user sojourn time increases when the number of users choosing the same BWP increases as users equally share limited amount of resources, which reduces the achieved throughput. Hence, an astute load balance of users among available configurations (single or multi-numerology) must be attained.

This cost function is later used in both the centralized and distributed approaches, which are detailed in subsequent sections.

## 4. Distributed Approach: Congestion Game

Users connected to multiple slices strive to choose between the multi-numerology (MN) and single-numerology (SN) BWP configurations in order to curb their energy consumption without sacrificing their QoS. Thus, the problem can be modeled as a non-cooperative game where the players are autonomous UEs competing over limited radio resources, which are the PRBs available in the BWPs. The corresponding non-cooperative game J is presented as follows:P is the set of players which are none other than the UEs connected to multiple slices.The set of strategies is S={MN,SN} where MN and SN designate the multi-numerology and single-numerology BWP configurations, respectively. Additionally, $v_{u}$ is the strategy vector of UE *u* which is composed of binary variables $v_{u,c}$. The latter is equal to 1 when UE *u* chooses the BWP configuration *c*. Therefore, the strategy profile is $v={\left(v_{u}\right)}_{u\in P}\in S$. Also, the space of all profiles is denoted by $S=S_{1}\times S_{2}\ldots \times S_{U}$.$\left\{C_{1},C_{2},\ldots ,C_{U}\right\}$ denotes the set of cost functions based on Equation (Equation 4) which assess the players’ gain after their strategy selection.

In J, the Nash Equilibrium (NE) is sought. In fact, the NE is an equilibrium state where every player *u* would select an optimal strategy in response to other players’ strategies and where no further benefit is acquired after deviating from this strategy unilaterally.

The game J is guaranteed to have a mixed NE, since it is a finite game. With mixed NEs, players select their optimal strategy following a probability distribution which may be inconvenient. However, thanks to the Finite Improvement Path property which J possesses similarly to the work in [6], the existence of a Pure NE (PNE) is guaranteed [18]. Thus, we prove this property below.

**Proposition 1.** 
*The game J holds the Finite Improvement Path property.*


**Proof.** The cost function of J is player specific and non-decreasing in the number of players which select the same strategy. This is because the sojourn time $\frac{V_{u,s}}{Thr_{u,s,bwp}}$ is non-decreasing in the number of players that selected the same strategy as the throughput is inversely proportional to the number of users selecting the same BWP configuration. Thus, J is an unweighted congestion game. Moreover, this type of game has the FIP property only when two strategies can be selected according to [19]. J verifies this condition, since only two strategies may be chosen MN and SN.    □

Subsequently, games holding the FIP property converge to the PNE using simple Best-Response dynamics, as stated by [20], where each player in turn will choose the strategy minimizing its cost function in response to other players’ strategies until convergence, when the chosen strategy of each player is the same as in the previous round presented in the below Algorithm 1.

In the next section, the same problem is tackled in a centralized approach.
**Algorithm 1:** Distributed BWP Configuration Selection Best Response Dynamics Algorithm
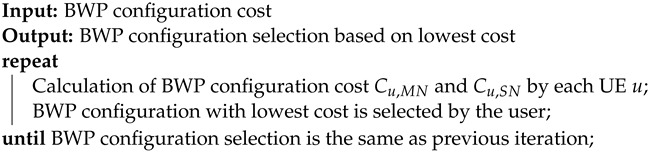


## 5. Centralized Approach: Optimization Problem

In the centralized approach, a central entity takes the decision of the BWP configuration selection for each user by solving an optimization problem corresponding to minimizing the total cost for all users.

### 5.1. The Objective Function

The optimization problem is the following:(6)$$\begin{array}{c} \;\min_{A1,A2}\sum_{u\in P}C_{u,MN}\left(A1\right)+\sum_{u\in P}C_{u,SN}\left(A2\right)\\ \mathrm{subject}\;\mathrm{to}:\\ {a_{1}}_{u},{a_{2}}_{u}\in \left\{0,1\right\},\forall u\in P\\ {a_{1}}_{u}+{a_{2}}_{u}=1,\forall u\in P \end{array}$$
where

$A1={\left({a_{1}}_{u}\right)}_{\forall u\in P}$ is the vector of binary decision variable ${a_{1}}_{u}$ which is equal to 1 when UE *u* has the MN strategy configured and equal to 0 otherwise.$A2={\left({a_{2}}_{u}\right)}_{\forall u\in P}$ is the vector of binary decision variable ${a_{2}}_{u}$ which is equal to 1 when UE *u* has the SN strategy configured and equal to 0 otherwise.$C_{u,MN}$ and $C_{u,SN}$ are the cost functions when UE *u* is configured with the MN and SN strategy, respectively, and are determined in the below Equations (Equation 7) and (Equation 8).

(7)$$\begin{array}{c} C_{u,MN}\left(A1\right)={a_{1}}_{u}\cdot \frac{\alpha_{u,eb}\cdot V_{u,eb}\cdot D1\cdot \left(\sum_{k\neq u}{a_{1}}_{k}+1\right)}{{TotPRB}_{BWP1}\cdot Bits\_per\_PRB_{u}}\\ +{a_{1}}_{u}\cdot \frac{\alpha_{u,uc}\cdot V_{u,uc}\cdot D2\cdot \left(\sum_{k\neq u}{a_{1}}_{k}+1\right)}{{TotPRB}_{BWP2}\cdot Bits\_per\_PRB_{u}}\\ +{a_{1}}_{u}\cdot \left(\theta_{MN}\cdot {SwDl}_{u}+\beta_{u}\cdot B_{MN}\right) \end{array}$$(8)$$\begin{array}{c} C_{u,SN}\left(A2\right)={a_{2}}_{u}\cdot \frac{\alpha_{u,eb}\cdot V_{u,eb}\cdot D3\cdot \left(\sum_{k\neq u}{a_{2}}_{k}+1\right)}{{TotPRB}_{BWP3}\cdot Bits\_per\_PRB_{u}}\\ +{a_{2}}_{u}\cdot \frac{\alpha_{u,uc}\cdot V_{u,uc}\cdot D3\cdot \left(\sum_{k\neq u}{a_{2}}_{k}+1\right)}{{TotPRB}_{BWP3}\cdot Bits\_per\_PRB_{u}}\\ +{a_{2}}_{u}\cdot \left(\beta_{u}\cdot B_{SN}\right) \end{array}$$
with D1, D2 and D3 representing the duration of the TTI when UE *u* is served by either BWP 1, 2 or 3, respectively.

Equations (Equation 7) and (Equation 8) are derived from Equations (Equation 1) and (Equation 3)–(Equation 5) after replacing each term. To note that the number of users attached to each BWP is determined by the sum of the binary decision variables which leads to $N_{users,BWP1}=N_{users,BWP2}=\sum_{k\neq u}{a_{1}}_{k}+1$ and $N_{users,BWP3}=\sum_{k\neq u}{a_{2}}_{k}+1$. Therefore, the aim of this optimization problem is to minimize the cost function of all users globally by choosing for every user the adequate binary variables ${a_{1}}_{u}$ and ${a_{2}}_{u}$. Note that the constraint ${a_{1}}_{u}+{a_{2}}_{u}=1,\forall u\in P$ is added to limit the user’s selection to only one strategy.

The optimization problem defined in Equation (Equation 6) is non-convex, since it is an integer (binary variables) non-linear problem due to the product of the decision variables with ${a_{1}}_{u}\cdot \left(\sum_{k\neq u}{a_{1}}_{k}+1\right)$ and ${a_{2}}_{u}\cdot \left(\sum_{k\neq u}{a_{2}}_{k}+1\right)$. Therefore, an exhaustive search may be used to solve this problem. With exhaustive search, all possible combinations for user distributions among the BWPs are assessed to choose the one with the lowest cost function, as seen in the below Algorithm 2. Nonetheless, this approach is highly time consuming with a complexity of $2^{N_{users}}$. For this reason, we transform the non-linear integer optimization problem in (Equation 6) to a linear one by replacing the multiplicative decision variables in what follows.
**Algorithm 2:** BWP Configuration Selection Exhaustive Search Algorithm
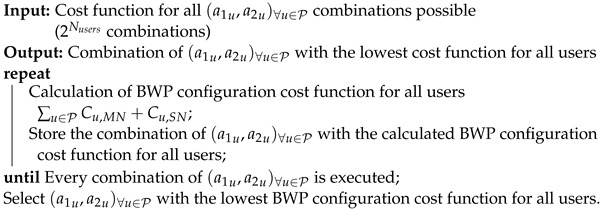


### 5.2. Integer Linear Programming Formulation

The non-linear terms ${a_{1}}_{u}\cdot {a_{1}}_{k}$ and ${a_{2}}_{u}\cdot {a_{2}}_{k}$ from (Equation 6) are replaced by the linear decision variables ${\pi_{1}}_{u,k}$ and ${\pi_{2}}_{u,k}$. Additionally, inequality constraints are added to ensure that the new variables are behaving the same way as the replaced non-linear terms. Also, ${\pi_{1}}_{u,u}$ is none other than ${a_{1}}_{u}$ since ${\pi_{1}}_{u,u}={a_{1}}_{u}\cdot {a_{1}}_{u}={a_{1}}_{u}$ since ${a_{1}}_{u}$ is binary. The same applies for ${\pi_{2}}_{u,u}$. Therefore, our Integer Linear Programming (ILP) problem is formulated as follows:(9)$$\begin{array}{cc} \; & \min_{\Pi 1,\Pi 2}\sum_{u\in P}C_{u,MN}\left(\Pi 1\right)+\sum_{u\in P}C_{u,SN}\left(\Pi 2\right)\\ \; & \mathrm{subject}\;\mathrm{to}:\\ \; & {\pi_{1}}_{u,u},{\pi_{2}}_{u,u}\in \left\{0,1\right\},\forall u\in P\\ \; & {\pi_{1}}_{u,u}+{\pi_{2}}_{u,u}=1,\forall u\in P\\ \; & {\pi_{1}}_{u,k}-{\pi_{1}}_{u,u}\leq 0,\forall u,k\in P\\ \; & {\pi_{2}}_{u,k}-{\pi_{2}}_{u,u}\leq 0,\forall u,k\in P\\ \; & {\pi_{1}}_{u,k}-{\pi_{1}}_{k,k}\leq 0,\forall u,k\in P\\ \; & {\pi_{2}}_{u,k}-{\pi_{2}}_{k,k}\leq 0,\forall u,k\in P\\ \; & {\pi_{1}}_{u,u}+{\pi_{1}}_{k,k}-{\pi_{1}}_{u,k}\leq 1,\forall u,k\in P\\ \; & {\pi_{2}}_{u,u}+{\pi_{2}}_{k,k}-{\pi_{2}}_{u,k}\leq 1,\forall u,k\in P \end{array}$$
where

$\Pi 1={\left({\pi_{1}}_{u,k}\right)}_{\forall u,k\in P}$ is the vector of binary decision variable ${\pi_{1}}_{u,k}$, which is equal to 1 when both users *u* and *k* have the MN strategy configured and equal to 0 otherwise.$\Pi 2={\left({\pi_{2}}_{u,k}\right)}_{\forall u,k\in P}$ is the vector of binary decision variable ${\pi_{2}}_{u,k}$, which is equal to 1 when both users *u* and *k* have the SN strategy configured and equal to 0 otherwise.${\pi_{1}}_{u,u}$ and ${\pi_{2}}_{u,u}$ are the same as ${a_{1}}_{u}$ and ${a_{2}}_{u}$ respectively.The first two constraints replace the ones defined in the previous problem, and the additional ones ensure that the new decision variables behave correctly with ${\pi_{1}}_{u,k}={a_{1}}_{u}\cdot {a_{1}}_{k}$ and ${\pi_{2}}_{u,k}={a_{2}}_{u}\cdot {a_{2}}_{k}$.$C_{u,MN}$ and $C_{u,SN}$ are the same cost functions when UE *u* is configured with the MN and SN strategy, respectively, but taking into account the new decision variables as shown in the below Equations (Equation 10) and (Equation 11).
(10)$$\begin{array}{c} C_{u,MN}\left(\Pi 1\right)=\frac{\alpha_{u,eb}\cdot V_{u,eb}\cdot D1\cdot \sum_{k\in P}{\pi_{1}}_{u,k}}{{TotPRB}_{BWP1}\cdot Bits\_per\_PRB_{u}}\\ +\frac{\alpha_{u,uc}\cdot V_{u,uc}\cdot D2\cdot \sum_{k\in P}{\pi_{1}}_{u,k}}{{TotPRB}_{BWP2}\cdot Bits\_per\_PRB_{u}}\\ +{\pi_{1}}_{u,u}\cdot \left(\theta_{MN}\cdot {SwDl}_{u}+\beta_{u}\cdot B_{MN}\right) \end{array}$$
(11)$$\begin{array}{c} C_{u,SN}\left(\Pi 2\right)=\frac{\alpha_{u,eb}\cdot V_{u,eb}\cdot D3\cdot \sum_{k\in P}{\pi_{2}}_{u,k}}{{TotPRB}_{BWP3}\cdot Bits\_per\_PRB_{u}}\\ +\frac{\alpha_{u,uc}\cdot V_{u,uc}\cdot D3\cdot \sum_{k\in P}{\pi_{2}}_{u,k}}{{TotPRB}_{BWP3}\cdot Bits\_per\_PRB_{u}}\\ +{\pi_{2}}_{u,u}\cdot \left(\beta_{u}\cdot B_{SN}\right) \end{array}$$

Therefore, this ILP problem can be solved with CPLEX solver, achieving the same optimal result as the exhaustive search algorithm but much more swiftly.

## 6. Performance Evaluation

We used Python for simulations to compare both approaches against each other and against the legacy scheme where all users are configured with a single BWP for both services with a band $B_{legacy}$. CPLEX solver was used to solve the ILP problem in the centralized approach. Simulations were run several times (≈100 times) with a different number of users each time, and the same results were observed regardless of the number of users. Therefore, we start by putting the focus on the scenario with $N_{users}=20$. Afterwards, we display results for $N_{users}=30$. The simulation parameters from the system model are shown in Table 1.

As for the results, we display the energy efficiency and total throughput Cumulative Distribution Functions (CDFs) for all users and the users’ overall sojourn time in the network CDF for all users. The energy efficiency of UE *u* selecting the strategy MN is computed as follows:
(12)$${EE}_{u,MN}=\frac{Thr_{u,eb}+Thr_{u,uc}}{P_{BWP1}+P_{BWP2}}$$
where

${EE}_{u,MN}$ is the energy efficiency of UE *u* selecting the MN strategy.$P_{bwp}$ is the UE power consumption for scanning BWP bwp.

When UE *u* selects the SN strategy, its energy efficiency is ${EE}_{u,SN}=\frac{Thr_{u}}{P_{BWP3}}$.

As for the overall sojourn time of UE *u*, it is $ST_{u,c}=\sum_{s\in \{eb,uc\}}DST_{u,s,c}+\theta_{c}\cdot SwDl_{u}$. As for the overall sojourn time of UE *u*, it is calculated as follows:(13)$$ST_{u,MN}=\sum_{s\in \{eb,uc\}}\frac{V_{u,s}}{Thr_{u,s}}+SwDl_{u}$$
(14)$$ST_{u,SN}=\frac{\sum_{s\in \{eb,uc\}}V_{u,s}}{Thr_{u}}$$

Figure 2 displays the number of users associated with each strategy using the exhaustive search algorithm for 20 users. Results show that a higher number of users prefer the single-numerology scheme, as it is more adapted to these users from an energy-efficiency perspective, since users scan a single and narrower BWP without BWP switching. Additionally, the single-numerology strategy is mainly selected for users with a low battery level on average, whereas the nulti-numerology scheme is chosen for users with a high battery level on average. This is due to the fact that users with a low battery level require more energy savings, which explains why they favor the single-numerology scheme.

Figure 3 represents the users’ energy-efficiency CDF for the centralized, distributed and legacy schemes for 20 users. It can be concluded that both proposed solutions—centralized and distributed—realize a higher energy efficiency than the legacy scheme thanks to the astute cost function that helped strike a nice balance between energy efficiency and QoS. Moreover, the centralized scheme performs slightly better than the distributed scheme.

Figure 4 represents the users’ total throughput CDF for the centralized, distributed and legacy schemes for 20 users. It is clear that both proposed solutions achieve higher throughput values than the legacy scheme thanks to the load balancing of users between the BWP configurations and thanks to the devised cost function that aims to increase energy efficiency, which leads to an increase of the users’ throughput.

Figure 5 shows the users’ sojourn time CDF for the centralized, distributed and legacy schemes for 20 users. The three schemes realize very close performances, which means that the proposed solutions aim to reduce the network’s sojourn time to a minimum.

As the number of users increases, the number of users with the MN BWP configuration is also increased to ensure a balance between both options with an inclination toward the SN BWP configuration. This can be seen in Figure 6, which represents the user distribution among strategies for 30 users.

As for the energy-efficiency CDF, Figure 7 displays it for 30 users where we can see that both solutions still provide better energy efficiency than the legacy scheme. However, the performance of these solutions becomes closer to the legacy scheme performance as the number of users increases, since the network becomes saturated. For the sojourn time CDF for 30 users, the same results are observed as the scenario of 20 users.

Figure 8 represents the users’ total throughput CDF for the centralized, distributed and legacy schemes for 30 users. Similar to the energy-efficiency results, we have lower throughput values with the increasing number of users. Nonetheless, higher throughput values are observed with the proposed solutions.

Therefore, we deduce that the proposed solutions (centralized and distributed) achieve better energy efficiency, which in turn helps with improving the users’ battery lifespan while ensuring the same sojourn time as the legacy scheme.

### 6.1. The Price of Anarchy

To adequately compare the centralized and distributed approaches, we have recourse to the well-known Price of Anarchy (PoA), which is computed as follows:(15)$$\mathrm{PoA}=\frac{{GlobalCost}_{optimal}}{{GlobalCost}_{NE}}$$

Therefore, it is the ratio of the sum of all users’ costs obtained with the optimal centralized approach and the sum of all users’ costs at NE for the distributed approach. The PoA is between 0 and 1, and the higher its value, the closer the performances of the distributed approach to the optimal centralized one.

Figure 9 displays the PoA as a function of the number of users. As can be seen, the PoA is ≈1 regardless of the number of users. This explains why the distributed approach attains very close performances to the centralized one.

As for the convergence time of each approach, it is represented in Figure 10 for the distributed approach with the best response algorithm and the centralized approach with CPLEX solver for the ILP problem.

The complexity of Algorithm 1 is $O(N_{users}\times N_{strategies}\times N_{iterations}=2)$ where $N_{users}$ is the number of users, $N_{strategies}$ is the number of strategies that can be selected which is equal to 2 (MN and SN strategies), and $N_{iterations}$ is the number of iterations until convergence is reached, which is also equal to 2 iterations. In fact, the algorithm duration is ≈0.01 s regardless of the number of users, which is very swift to meet the stringent delay requirements. Nonetheless, the centralized solution with the ILP problem takes a slightly higher computation time of ≈10 s when $N_{users}\leq 20$ but starts to increase exponentially when $N_{users}>20$. In the latter case, the convergence time is too long, which will result in failing to respect the tolerated latency. However, the learning of the optimal choice of strategy may be learned offline prior to the arrival of users in order to make fast decisions. Also, it is important to note that for the centralized approach, the ILP problem solver CPLEX remains much faster than the exhaustive search algorithm where the convergence time is ≈300 s for $N_{users}=20$.

We conclude that the distributed approach is able to achieve almost optimal results with a much lesser computational time compared to the optimal centralized approach where the convergence time becomes significantly important with an increasing number of users.

### 6.2. Results Highlight

Based on these results, we can draw the following conclusions:Most users are likely to select the single-numerology BWP configuration, since it provides a reduced power consumption. The latter increases with the frequency of BWP scanning (Figure 2 and Figure 6).Both approaches (centralized and distributed) provide better energy efficiency and throughput while maintaining the UE’s network sojourn time to a minimum thanks to the well-defined cost function (Figure 3, Figure 4, Figure 5, Figure 7 and Figure 8).As the number of users increase, the performances become closer to the legacy scheme since the network becomes saturated (Figure 6, Figure 7 and Figure 8).The distributed and centralized approaches realize almost the same performances as can be explained by the PoA analysis (Figure 9).The distributed approach has the best performance in terms of convergence time while achieving almost the same QoS performance as the centralized approach (Figure 9 and Figure 10). Hence, the distributed approach should be adopted for implementation.

## 7. Conclusions

This paper proposes a savvy and flexible scheme to select an appropriate BWP configuration for users connected to multiple slices that help such users reduce their energy consumption without hindering their QoS. The BWP configuration selection between multi-numerology and single-numerology BWPs is assessed for each user depending on multiple factors, including battery level and QoS satisfaction. Two approaches are adopted. The first approach is a centralized one based on a global optimization problem where a central entity minimizes the total cost of users by selecting for each user the most adequate BWP configuration. The second approach is a distributed one based on non-cooperative game theory, where each user selects autonomously the BWP configuration that minimizes its own cost. Extensive simulations prove the efficiency of our devised scheme against the static legacy scheme, and our evaluation of the price of anarchy proves the precedence of the distributed approach over the centralized one, as it combines fast convergence and near optimal performances.

For future work, we intend to take into account additional services for users connected to multiple slices.

## Figures and Tables

**Figure 1 sensors-24-01281-f001:**
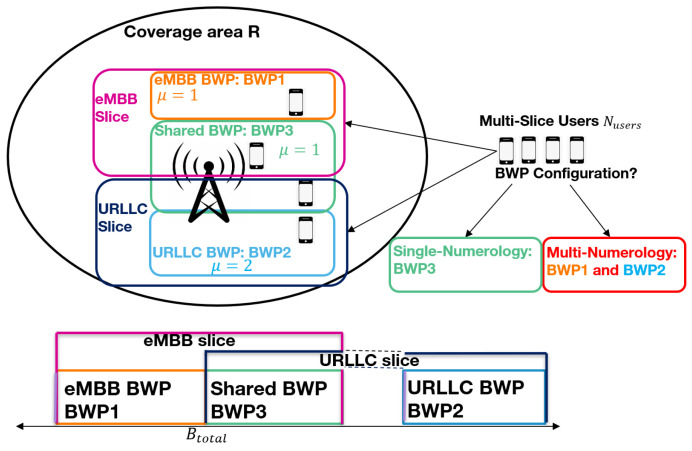
System model.

**Figure 2 sensors-24-01281-f002:**
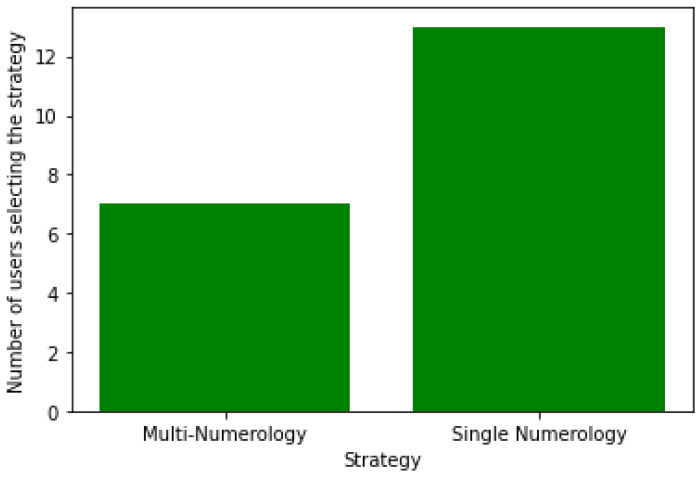
User distribution among strategies with an exhaustive research algorithm for 20 users.

**Figure 3 sensors-24-01281-f003:**
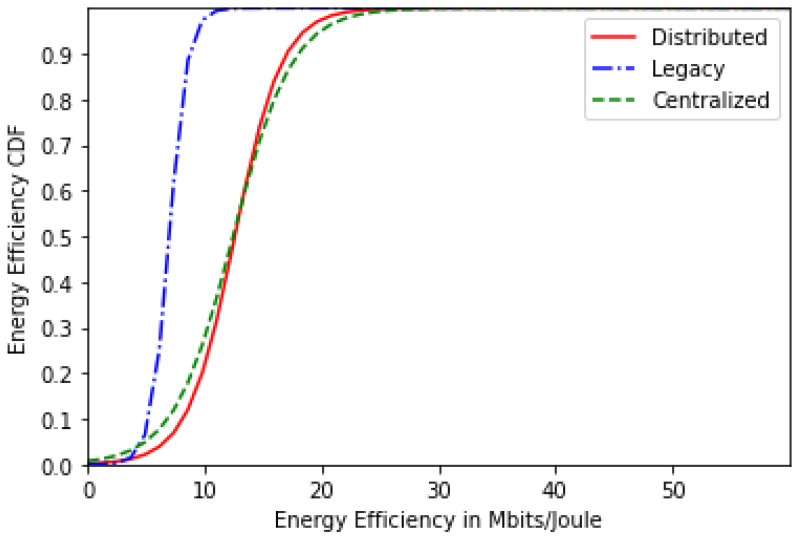
Energy-efficiency CDF for 20 users.

**Figure 4 sensors-24-01281-f004:**
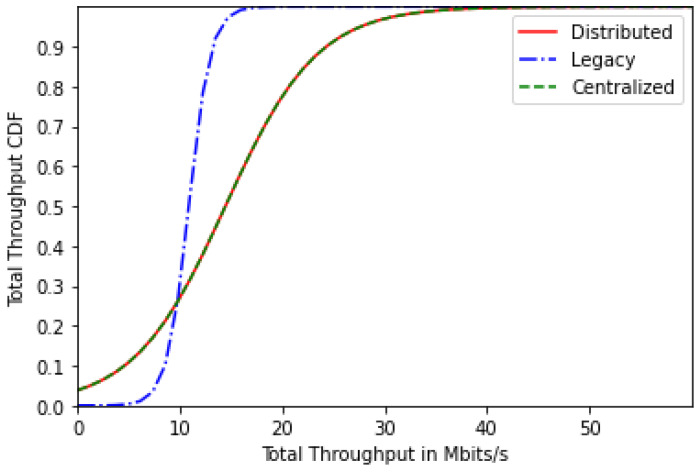
Total throughput CDF for 20 users.

**Figure 5 sensors-24-01281-f005:**
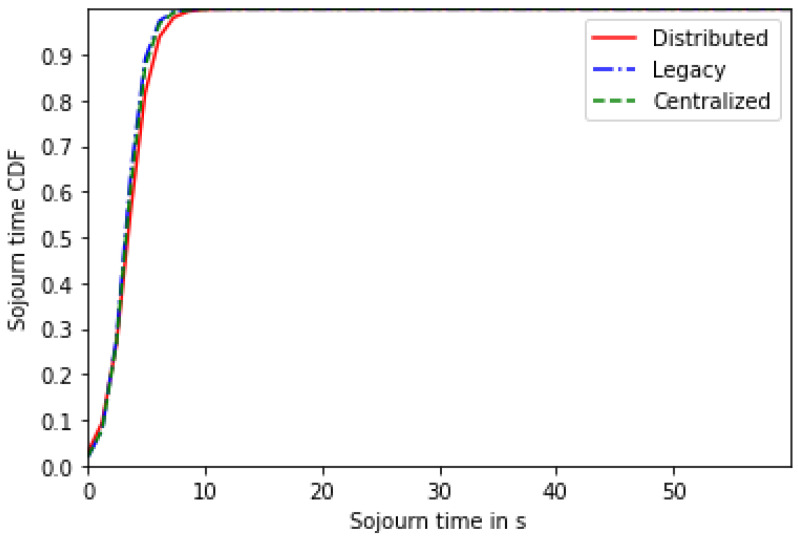
Sojourn time CDF for 20 users.

**Figure 6 sensors-24-01281-f006:**
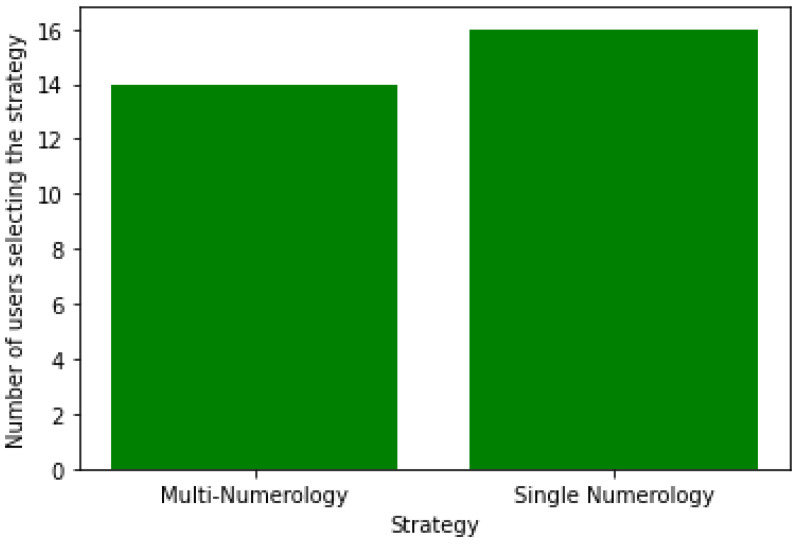
User distribution among strategies with centralized solution algorithm for 30 users.

**Figure 7 sensors-24-01281-f007:**
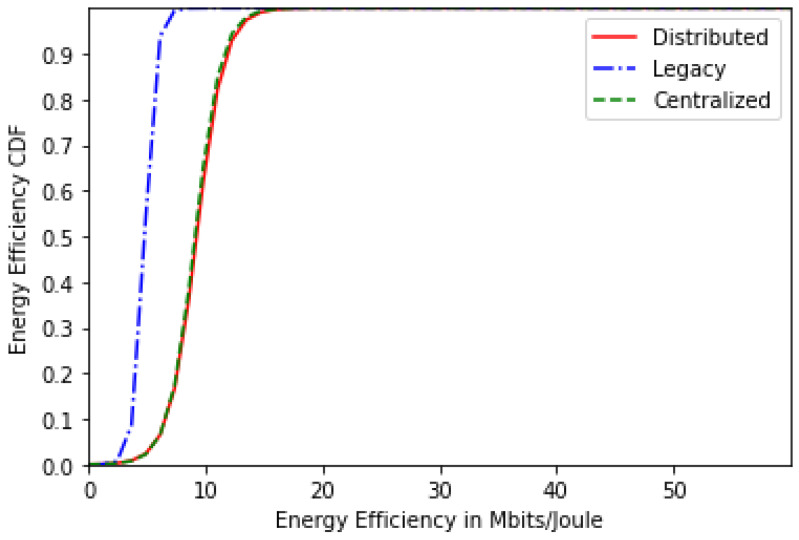
Energy-efficiency CDF for 30 users.

**Figure 8 sensors-24-01281-f008:**
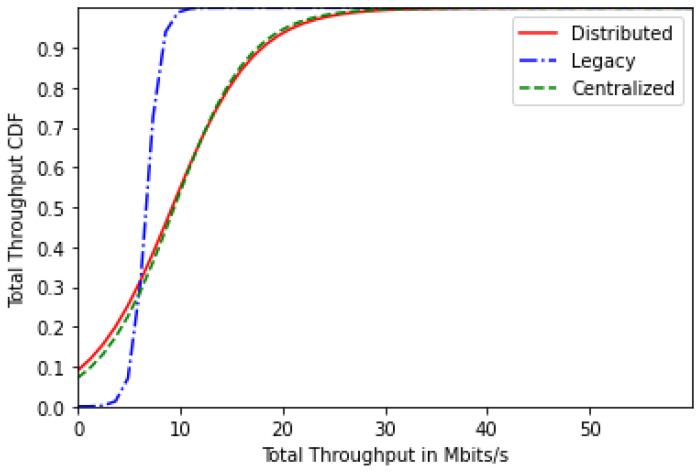
Total throughput CDF for 30 users.

**Figure 9 sensors-24-01281-f009:**
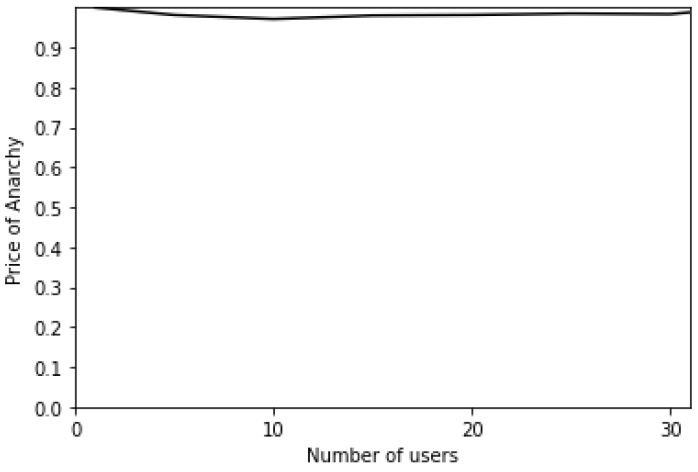
Price of Anarchy as a function of the number of users.

**Figure 10 sensors-24-01281-f010:**
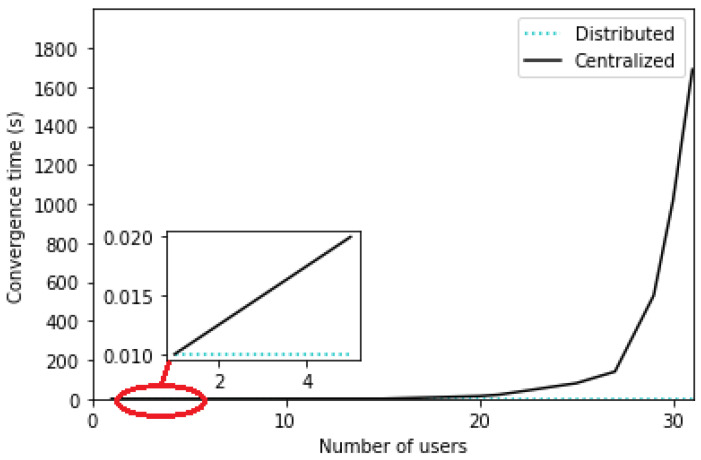
Convergence time as a function of the number of users.

**Table 1 sensors-24-01281-t001:** Simulation parameters.

Parameter	Description	Value
*R*	Coverage area	300 m
$V_{URLLC}$	URLLC volume of data	2–15 Mbits
$V_{eMBB}$	eMBB volume of data	5–50 Mbits
$SwDl_{u}$	User’s BWP switching delay	1 ms [5]
$B_{total}$	Operator band	100 MHz
$B_{eMBB}$	eMBB BWP band	40 MHz
$B_{URLLC}$	URLLC BWP band	20 MHz
$B_{mixed}$	Mixed BWP band	30 MHz
$B_{legacy}$	Legacy BWP band	60 MHz
$P_{BWP1}$	Power consumption for scanning BWP 1 (eMBB)	1000 mW
$P_{BWP2}$	Power consumption for scanning BWP 2 (URLLC)	500 mW
$P_{BWP3}$	Power consumption for scanning BWP 3 (mixed)	750 mW
$P_{legacy}$	Power consumption for scanning legacy BWP	1500 mW
$\alpha_{u,eMBB}$	Normalizing factor for cost function	1
$\alpha_{u,URLLC}$	Normalizing factor for cost function	2
$\beta_{u}$	Normalizing factor for cost function	0.001–0.099

## Data Availability

Data will be made available upon request.

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
