# Peer review of "Energy-Efficient BWP Configuration for Multi-Slice Users"

_sensors, 2024, doi:10.3390/s24041281_

Round 1
Reviewer 1 Report
Comments and Suggestions for Authors
This paper considered two energy efficient Bandwidth Part selection solutions are proposed for users connected to multiple slices for 5G mobile networks. In the work centralized and user-centric distributed approaches were used with a view of taking into account the EU’s battery level. The paper also compared the performance of the two approaches. Intensive simulation results demonstrate the effciency of the proposition in terms of users’ energy efficiency and quality of service.
The work is timely and provide some useful results. However, some sections of the paper need to be reworked on as summarise as follows:
1. The review Section
I feel the review has not provided clear demarcation of the previous works. Aside this work considered users connected to multiple slices, I couldn’t see much difference with existing papers.
2. System Model
A diagram illustrating the approaches should be provided for more clarity
3. The results discussion is not adequate, the paper couldn’t draw important conclusions on to which of the configurations performed better.
Comments on the Quality of English LanguageThe paper is well written, only little editorial required.0
Author Response
Please see the attachment and the following authors' response.
This paper considered two energy efficient Bandwidth Part selection solutions are proposed for users connected
to multiple slices for 5G mobile networks. In the work centralized and user-centric distributed approaches
were used with a view of taking into account the EU’s battery level. The paper also compared the performance
of the two approaches. Intensive simulation results demonstrate the effciency of the proposition in terms of
users’ energy efficiency and quality of service.
The work is timely and provide some useful results. However, some sections of the paper need to be reworked
on as summarise as follows:
RC: 1. The review Section. I feel the review has not provided clear demarcation of the previous works. Aside
this work considered users connected to multiple slices, I couldn’t see much difference with existing papers.
AR: We thank the reviewer for their assessment. In order to clarify the novelty of our work compared to other
existing works, we added further demarcation for each previous work in the Related Works section which are
highlighted in yellow in the revised manuscript. In fact, multi-slice users are not the sole novelty of our work
as other points are listed to underline our contributions as follows:
• Users connected to multiple slices are considered which are rarely tackled in the SOTA.
• The selection of a BWP configuration for multi-slice users is used to address the radio resource
allocation problem unlike other works from the literature focusing on the PRB allocation.
1
• This BWP configuration scheme is based on a novel concept where either the Multi-Numerology or
Single Numerology BWP configuration is selected for each multi-slice user while aiming to optimize
the users’ energy consumption and QoS.
• The energy efficiency aspect is considered in the BWP configuration selection process instead of using
existing mechanisms from the standards such as DRX and BWP inactivity timer to reduce users’ energy
consumption.
• A centralized and a distributed approach are proposed and are compared against each other and against
the legacy scheme.
This summary of our contributions compared to other works has been added in the end of the Related Works
section to highlight these demarcations.
RC: 2. System Model: A diagram illustrating the approaches should be provided for more clarity
AR: Thank you for this remark. Figure 1 representing the System Model has been added to the revised manuscript
for further clarification.
RC: 3. The results discussion is not adequate, the paper couldn’t draw important conclusions on to which of
the configurations performed better.
AR: The comments following the figures were adjusted and a subsection entitled "Results highlight" was added.
This subsection summarizes the important results and discusses the configuration with the best performance
as follows:
• Most users are likely to select the Single Numerology BWP configuration since it provides a reduced
power consumption. The latter increases with the frequency of BWP scanning (Figs. 2 and 6).
• Both approaches (centralized and distributed) provide better energy efficiency and throughput while
maintaining the UE’s network sojourn time to a minimum thanks to the well-defined cost function
(Figs. 3, 4, 5, 7, 8).
• As the number of users increase, the performances become closer to the legacy scheme since the
network becomes saturated (Figs. 6, 7, 8).
• The distributed and centralized approaches realize almost the same performances as can be explained
by the PoA analysis (Fig. 9).
• The distributed approach has the best performance in terms of convergence time while achieving almost
the same QoS performance as the centralized approach (Figs. 10,9). Hence, the distributed approach
should be adopted for implementation.

Reviewer 2 Report
Comments and Suggestions for Authors
This work studied Bandwidth Part (BWP) configuration for multi-slice users for improving energy efficiency and quality of service, which is a promising research direction. This work presented an integral linear programming centralized approach and a distributed game algorithm, and verified their efficiency by comparing a legacy solution. Before this work's publication, there are some issues needed to be addressed.
1) Newly published solutions need to be compared by experiments, to reliably confirm the efficiency and effectiveness of proposed methods.
2) The proposed approaches need to be detailed.
3) Experimental results need to be analysed deeply. The rationality of results should be discussed.
4) The abstract is too verbose, especially the background.
5) It will be better to provide diagrams when stating the problem and describing their proposed methods.
6) The integral linear programming centralized approach provides the best solution, but why sometimes it has poor performance compared with the distributed game algorithm?
Comments on the Quality of English LanguageSome sentences are less readable. For example, the length of the first sentence is almost 5 lines, making it too hard to be understood.
Author Response
Please see the attachment and the following authors' response:
This work studied Bandwidth Part (BWP) configuration for multi-slice users for improving energy efficiency
and quality of service, which is a promising research direction. This work presented an integral linear programming
centralized approach and a distributed game algorithm, and verified their efficiency by comparing
a legacy solution. Before this work’s publication, there are some issues needed to be addressed.
RC: 1) Newly published solutions need to be compared by experiments, to reliably confirm the efficiency and
effectiveness of proposed methods.
AR: We thank the reviewer for their assessment. Note that our solutions have been validated via extensive
simulations (repeated almost 100 times) with a different number of users each time. The same results were
observed regardless of the number of users which confirms the efficiency of our solution. This has been added
at the beginning of the performance evaluation section of the revised manuscript.
RC: 2) The proposed approaches need to be detailed.
AR: To provide more clarification about the contributions of our work and the proposed approaches, a summary of
our contributions was added at the end of the Related Works section to underline our proposed solutions. In
addition, the proposed approaches (centralized and distributed) are detailed in Sections 4 and 5 where the
problem is described as well as the mathematical tool to solve it.
RC: 3) Experimental results need to be analysed deeply. The rationality of results should be discussed.
AR: The most important conclusions drawn from the results analysis was added in a new subsection entitled
"Results Highlight" in the Performance Evaluation section in the revised manuscript. Each conclusion point
is linked to the figure where the result was analyzed.
RC: 4) The abstract is too verbose, especially the background. Some sentences are less readable. For example,
the length of the first sentence is almost 5 lines, making it too hard to be understood.
AR: The abstract and the first sentence were revised in the new version of the manuscript accordingly.
RC: 5) It will be better to provide diagrams when stating the problem and describing their proposed methods.
AR: Thank you for this comment. Figure 1 was added to display the system model and the problem to be solved
for better clarification.
RC: 6) The integral linear programming centralized approach provides the best solution, but why sometimes it
has poor performance compared with the distributed game algorithm?
AR: It is true that the centralized approach provides the best QoS performance. Nonetheless, its convergence time
increases exponentially with the number of users. Thus, it is better to implement the distributed approach
since it achieves near optimal performance as can be seen in the Price of Anarchy analysis with a much faster
convergence time. This has been added in the Results Highlight subsection. As for the QoS performance,
the centralized approach always slightly outperforms the distributed one. However, the mentioned poor
performance is due to the curve-fitting function that aligns the collected data and adjusts the obtained
results curve of each solution leading to a small dissimilarity between the curves. Although quite close, the
centralized approach always performs slightly better than the distributed approach in terms of QoS satisfaction
and energy efficiency as explained in the results highlight.

Reviewer 3 Report
Comments and Suggestions for Authors
The proposed work is very well organized, the developed system model is well defined, and the experimental results are very interesting. I didn’t find any grammatical issues, or detect plagiarism, from what I searched. Finally, I didn’t detect inappropriate self-citations by authors. Thus, in my opinion the manuscript should be published.
Prior this, I have some remarks that are as follows:
1. The sentence: “the PRB is considered to be composed of 12 subcarriers in the frequency domain and 14 OFDM symbols in the time domain which explains the displayed values “ is repeated in lines 144-145 and 148-149. Authors should review this.
2. Authors should mention for which frequency band of the URLLC-5G the proposed System Model that they use applies to.
3. In lines: 127 and 129 authors define BeMBB and BuRLLc without giving any detail about the factors γeMBB and δURLLC. Authors should mention more about these equations.
Author Response
Please see the attachment and the following authors' response:
The proposed work is very well organized, the developed system model is well defined, and the experimental
results are very interesting. I didn’t find any grammatical issues, or detect plagiarism, from what I searched.
Finally, I didn’t detect inappropriate self-citations by authors. Thus, in my opinion the manuscript should be
published.
Prior this, I have some remarks that are as follows:
RC: 1. The sentence: “the PRB is considered to be composed of 12 subcarriers in the frequency domain and 14
OFDM symbols in the time domain which explains the displayed values “ is repeated in lines 144-145 and
148-149. Authors should review this.
AR: We thank the reviewer for their review. The mentioned repetition was removed and addressed in the revised
manuscript.
RC: 2. Authors should mention for which frequency band of the URLLC-5G the proposed System Model that
they use applies to.
AR: Thank you for this remark. The following sentence was added in the System Model to indicate on which
frequency the gNB operates: "The gNB covers an area with radius R. Additionally, it operates on the
foperator = 3.5 GHz frequency (Frequency Range 1) in Time-Division Duplex (TDD) mode."
RC: 3. In lines: 127 and 129 authors define BeMBB and BuRLLc without giving any detail about the factors
γeMBB and δURLLC. Authors should mention more about these equations.
AR: These factors were added to indicate that we attribute a certain portion of the total band to each BWP and the
sum of these factors is less or equal to 1 meaning that the total band may not be totally utilized. However,
these factors are not relevant for the rest of the paper. For this reason, they were removed from the revised
manuscript and we simply added the condition BeMBB + BURLLC + Bmixed ≤ Btotal. Hence, the revised
part is as follows: "Additionally, three bandwidth parts are considered for each slice: the first BWP denoted
by BWP 1 consists of the band attributed to the eMBB slice and uses numerology 1 (the lower numerology)
with a band BeMBB MHz, the second BWP denoted by BWP 2 is the band attributed to the URLLC slice
using numerology 2 (a higher numerology) with a band BURLLC MHz. To note that users selecting the
multi-numerology BWP configuration will scan both these BWPs (BWP 1 and BWP 2) consecutively to
retrieve the data for each slice. The third BWP denoted by BWP 3 is the one shared between both slices using
numerology 1 with a band Bmixed with BeMBB + BURLLC + Bmixed ≤ Btotal."

Round 2
Reviewer 2 Report
Comments and Suggestions for Authors
No comment